# Heating Energy Consumption and Environmental Implications Due to the Change in Daily Habits in Residential Buildings Derived from COVID-19 Crisis: The Case of Barcelona, Spain

Marta Monzón-Chavarrías [1] , Silvia Guillén-Lambea [2,3] , Sergio García-Pérez [1] , Antonio Luis Montealegre-Gracia [4] and Jorge Sierra-Pérez [5,6,*]

1 Department of Architecture, University of Zaragoza, 50018 Zaragoza, Spain; monzonch@unizar.es (M.M.-C.); sgarciap@unizar.es (S.G.-P.)
2 University Center of Defense, University of Zaragoza, 50090 Zaragoza, Spain; sguillen@unizar.es
3 Thermal Engineering and Energy Systems Group (GITSE), Aragón Institute for Engineering Research (I3A), University of Zaragoza, 50018 Zaragoza, Spain
4 GEOFOREST-IUCA Research Group, Department of Geography, University of Zaragoza, 50009 Zaragoza, Spain; monteale@unizar.es
5 Department of Design and Manufacturing Engineering, EINA, University of Zaragoza, 50018 Zaragoza, Spain
6 Water and Environmental Health Research Group, University of Zaragoza, 50018 Zaragoza, Spain
* Correspondence: jsierra@unizar.es; Tel.: +34-976-761-905

**Abstract:** The COVID-19 crisis has changed daily habits and the time that people spend at home. It is expected that this change may have environmental implications because of buildings' heating energy demand. This paper studies the energy and environmental implications, from a Life Cycle Assessment (LCA) approach, due to these new daily habits in residential buildings at their current level of thermal insulation, and in different scenarios of thermal retrofit of their envelope. This study has a building-to-building approach by using Geographical Information Systems (GIS) for the residential housing stock in the case of Barcelona, Spain. The results show that a change in daily habits derived from the pandemic can increase the heating energy consumption and carbon dioxide emission in residential buildings by 182%. Retrofitting all buildings of Barcelona, according to conventional energy renovation instead of nearly Zero Energy Buildings (nZEB), will produce between $2.25 \times 10^7$ and $2.57 \times 10^7$ tons of carbon dioxide. Retrofitting the building stock using energy recovery is the option with better energy and emission savings, but also is the option with higher payback time for buildings built until 2007. The methodology presented can be applied in any city with sufficient cadastral data, and is considered optimal in the European context, as it goes for calculating the heating energy consumption.

**Keywords:** COVID-19; GIS; LCA; urban scale; user behaviour; nZEB renovation; energy retrofit

## 1. Introduction

Buildings are responsible for the 40% of energy consumption in Europe [1]. Almost 50% of the European Union's (EU) final energy consumption is used for heating and cooling in which 80% is used in buildings [2]. The European Union has established the requirement that all Member States (MS) should reduce the greenhouse gas (GHG) emissions from the building stock by at least 55% by 2030 and by at least 90% in 2050. To achieve this goal, it is necessary not only to reduce energy consumption in new buildings, but to retrofit the building stock. According to Europe's Buildings under the Microscope, the percentage of residential buildings erected before 1990 is 86% in Southern Europe, 83% in Central and Eastern Europe, and 81% in North-Western Europe [3]. The 85–95% of the buildings that exist today will still be standing in 2050 [4]. In Spain, 66% of building stock are residential buildings [5] and 56.3% of residential buildings are erected before 1980, i.e., before the first energy saving regulations for buildings were approved [6].

The European Parliament has promoted different directives for energy efficiency in the buildings sector. The Energy Performance of Buildings Directive (EPBD) 2010/31/EU [1] establishes as a requirement that all new buildings must be nearly Zero Energy Buildings (nZEB) from the end of 2020, and establish minimum energy performance requirements for a major renovation. However, it is expected that the tendency of most of the European countries will be to renovate their old buildings better than to erect new buildings. Hence, the European Strategies should be focused on renovating housing stock. Retrofitting the housing stock not only has energy benefits, but also can improve thermal comfort and decrease the impact of energy poverty, which is estimated to affect more than 34 million households in the EU [7].

The COVID-19 health crisis has changed dramatically the patterns of energy use in the countries. In 2020, many countries established restrictive lockdown and movement restriction measures. Some papers study the energy implications during this time. According to Yusup et al. [8], during the first semester of 2020, the global energy demand was reduced by 3.7%. Santiago et al. [9] concluded that, during lockdown in Spain, the electricity demand has decreased by 13.5% and $CO_2$ emissions have decreased by 32.61% compared to 2019. Similar to Italy, the pandemic caused a reduction in electricity consumption by up to 37% compared to the same time as the previous year [10]. In the case of China, its $CO_2$ emission decreased 11% during the first quarter of 2020 related to COVID-19 mitigation measures [11]. Other authors study the energy transition opportunity due to innovative technologies with increased usage due to COVID-19 [12], which involves the transition to renewable energy [13–15].

The COVID-19 health crisis has led to an increase in the time people spend inside their homes. Many countries established very restrictive measures of lockdown for a certain time during 2020, and, subsequently, it has been recommended to stay at home and favour teleworking to avoid contagion. The pandemic has brought into sharper focus our buildings. The home has been the focal point of daily life for millions of Europeans: an office for those teleworking, a nursery or classroom for children and pupils, and a hub for online shopping or downloading entertainment for many [4]. The time people spend in the office, schools, and leisure time has been largely replaced by spending time into homes. During this situation, where most people spend the most time inside their dwellings, the total energy consumption of buildings will be influenced mostly by energy consumption of residential buildings. Some of the effects of the pandemic may continue in the longer term, creating new demands on our buildings and their energy and resource profile, further adding to the need to renovate them deeply and on a massive scale [4]. As Europe seeks to overcome the COVID-19 crisis, renovation offers a unique opportunity to rethink, redesign, and modernize our buildings to make them fit for a greener and digital society and sustain economic recovery [4].

The home office has suddenly experienced a rebound as a result of the measures to protect citizens from the coronavirus disease [16]. Working from home has increased slowly in recent years in European countries before the COVID-19 outbreak. In 2019, 14.1% of employed persons in the Netherlands and Finland always worked from homeand 30% in these countries and Sweden worked from home regularly [17]. That year in Spain, only 4.8% of employed persons worked from home. However, this percentage is expected to increase due to European recommendations [18] and the recent regulation approved in Spain [19]. Some authors detect the tendency to telework and study its implications. The Abulibdeh study the transition to telework during the COVID-19 pandemic and check the continuity of teleworking practices after the pandemic and its benefits for the economy and inequality [20]. Barnes and Beaunoyer et al. identify teleworking and distance learning as an opportunity in information management research in the post-COVID world [21,22].

The energy retrofit of residential buildings is more important in this situation to avoid increasing energy consumption, $CO_2$ emission, and energy poverty.

Applying circularity principles to building renovation will reduce materials-related greenhouse gas emissions for buildings. One of the key principles for building renovation

toward 2030 and 2050, according to the European Commission, is minimizing the footprint of the building that requires resource efficiency and circularity combined with turning parts of the construction sector into carbon sink [4]. Circular economy actions can lead to reductions of up to 60% in the materials-related greenhouse gases emitted across the life-cycles of buildings. Investing in buildings can also inject a much-needed stimulus in the construction ecosystem and the broader economy [4].

The change in time people spend at home can increase the energy consumption and Greenhouse Gases (GHG) emission of residential buildings more than is expected. Studying this potential with LCA and economic implications at city scale will allow policy makers to design local strategies to promote energy retrofitting. Some recent articles that have been analysed in previous paragraphs studied the effects of lockdown and other restrictive measures during the 2020 crisis due to COVID-19. However, these data are about global energy during a specific period in the past time.

Some authors use bottom-up methodologies and Geographical Information systems (GIS) data to study the current energy performance at urban scale. Lorenzo et al. [23] mapping the primary energy consumption and GHG emissions of the existing buildings in Valencia (Spain) using the existing energy efficiency certificates. Torabi et al. [24] and Mastrucci et al. [25] estimate the energy demand and energy consumption of city residential building stocks for heating space, using a GIS-statistical methodology. Tooke et al. [26] use Light Detection and Ranging (LIDAR) data and statistical methodology to estimate and mapping the energy demand in a residential neighbourhood in Vancouver (Canada) studying envelope characteristics and solar gains.

Other authors study the energy saving potential of retrofitting existing buildings at an urban scale. García-Ballano et al. [27] use GIS and cadastral data to evaluate the potential energy savings generated by the energy retrofit of the housing stock of Zaragoza (Spain). However, this study does not consider the environmental nor economic implications. García-Pérez et al. [28–30] study the environmental benefits at urban scale of using natural insulation in buildings retrofitting in the case of Barcelona (Spain). However, this paper does not show energy consumption savings nor economic costs. Mastrucci et al. [31] propose a method to determine the energy demand and carbon footprint reduction potential for renovating the existing housing stock of Esch-sur-Alzette (Luxembourg). To calculate the energy demand, this paper uses a simplified equation. However, this study does not consider the new change in daily life habits because of the COVID-19 disease nor the economic cost.

This paper presents a methodology, which analyses at urban scale the influence of heating energy consumption due to changes in daily life habits in residential buildings through different energy retrofit scenarios. This methodology allows us to observe the influence on changes in daily habits due to the COVID-19 crisis in terms of the potential energy savings consumption, environmental implications, and economic cost. The study uses a building-by-building approach and by means of a bottom-up methodology extrapolates the unitary result to urban scale. The methodology is applied to the case in the city of Barcelona. The implications at the city scale refer to the existing residential building stock in Barcelona built until 2015.

## 2. Materials and Methods

The research methodology includes the next steps, which are developed in the following sections and are summarized in Figure 1.

a. Characterization of the buildings of Barcelona. The residential buildings are classified according to the year of construction.
b. Dwellings selection. Representative dwelling types are selected depending on the type of building and the position of the dwelling in the building.
c. Inhabitant change in daily habits.
d. Scenarios definition. The three preceding classifications will allow defining the scenarios together with three proposed renovation strategies.

e.　Energy consumption. The constructive parameters for each scenario are defined to estimate the heating energy demand and consumption for each defined scenario.
f.　LCA will be performed for each defined scenario.
g.　Cost estimations.
h.　Extrapolation from the building to the urban level.

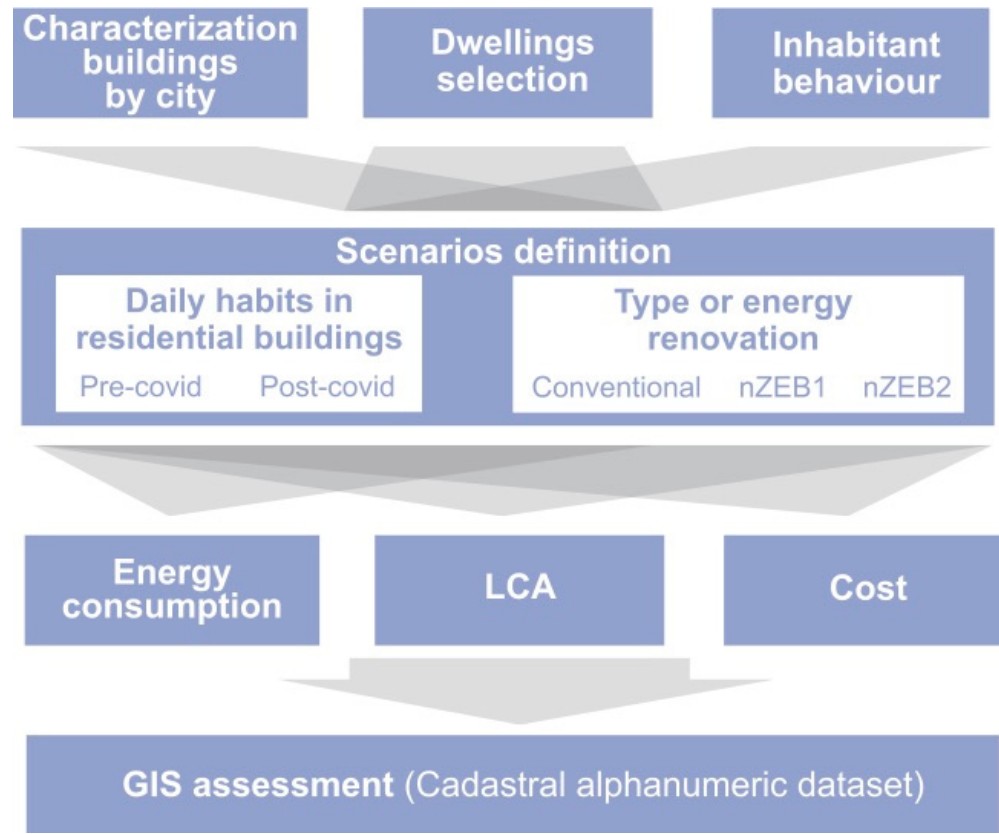

**Figure 1.** Methodology outline.

### 2.1. Characterization of the Buildings of Barcelona City

The air volume of residential buildings to be conditioned in the city of Barcelona is determined using GIS data and Spanish Cadastral database, which is the main public administration dataset of building stock information in Spain. Cadastral data contains open information related to each cadastral reference as type of use, year of construction, built area, and floors number, among others.

Constructive solutions are usually linked to the year of construction [5]. Therefore, this study determines the age of construction of the buildings using cadastral data, according to three groups outlined below.

- Buildings built before 1981
- Buildings built between 1981 and 2007
- Buildings built after 2008

That group of years of construction are used in other studies [32,33] because, in Spain, the first regulation on energy performance of new buildings was applied since 1980 [33], and the second one was approved in 2006 [34]. Based on the period of construction, the constructive solutions are chosen as explained in Section 2.4.

### 2.2. Dwelling Selection

The guide n°12: "Simplify option. Houses. Calculation report" produced by the Institute for Energy Diversification and Saving (IDAE) [35] has substantial data of building

characteristics, the number of dwellings, and their average area in Spain. The most common dwelling area is between 76 m$^2$ and 90 m$^2$. The representative dwelling has a living room, a kitchen, three double bedrooms, and two bathrooms. The final report of the SPAHOUSEC project (Analysis of the Energy Consumption in Spanish Households-IDAE) [36] gives information about the housing census and energy consumption. It can be deduced from this report that more than 70% of Spanish homes are in blocks and their average surface area is 86.5 m$^2$. The selected dwelling is in a housing block and has a representative size and layout for a typical family. It has a kitchen, a living room, three bedrooms, and two bathrooms. The net area is 81.15 m$^2$ and the ceiling height is 2.5 m.

Finally, two dwellings were selected depending on their position on the apartment block. The first one is located at mid-floor (MF) and the second one is located on the top-floor (TF) of a multi-family building. The distribution, surface, and orientation of the dwelling is used in previous studies [37]. Regarding its orientation, the dwelling has windows on the north facade in the living room and in the double bedroom, and on the south facade in the kitchen, bathroom, and two bedrooms. Only the hall and toilet have no exterior windows. The following requirements for the selected dwellings were selected.

- U-values of the envelope [35]
- 23.6% of the windows and external wall areas versus the net area of the dwelling [37]

### 2.3. Inhabitant Change in Daily Habits

User behaviour has a relevant impact in the energy performance of buildings [38], mainly related to two parameters, first, the comfort parameters definition and, second, the increase of internal gains due to increased overall time at home for all residents. Regarding the comfort parameters, it is evident that the increase in the time of presence of the residents at home leads to an increase, or at least a maintenance of the comfort temperature in the home during the entire time they stay at home.

The main internal loads in residential buildings are due to the lighting, equipment, and home appliances and the metabolic activity of the occupants. Accordingly, not only does the presence of people increase the internal loads in the home, but a greater use of equipment (computers, ovens, kitchen, television . . . ) takes on special relevance. The internal comfort and gain parameters have been established for two states known as the pre-pandemic and the post-pandemic states.

For the pre-pandemic state, the temperature set values for heating in winter and the internal gains are based on the technical code of the Spanish building [39,40], which provides a temperature of 17 °C (from 0:00 to 7:00 and from 23:00 to 23:59) and 20 °C (from 7:00 to 22:59). The heat generation according to different degrees of activity follows the values detailed in ISO 7730: 2005 [41]. Three people are living in the house, one with a nominal load 60 W sensible and 40 W latent and two with 65 W sensible and 55 W latent. For lighting and equipment, a nominal load of 5 W/m$^2$ has been considered. For the pre-pandemic state, the nominal loads are multiplied by a coefficient depending on the time of day related to the occupancy, as indicated in Table 1 [42,43].

**Table 1.** Coefficients of internal loads applied in the model depending on the time of day (Source: [42,43]).

| | Time of Day-Working Day | | | | | Time of Day-Weekend | | | | |
|---|---|---|---|---|---|---|---|---|---|---|
| | 0–7 | 7–13 | 13–15 | 15–20 | 20–24 | 0–9 | 9–12 | 12–17 | 17–22 | 22–24 |
| Occupancy | 0.50 | 0.25 | 1.00 | 0.50 | 0.75 | 0.50 | 0.25 | 2.00 | 0.50 | 1.00 |
| Lighting and equipment | 0.00 | 0.50 | 0.00 | 1.00 | 0.50 | 0.00 | 0.50 | 1.50 | 1.00 | 0.50 |

For the post-pandemic state, the temperature value for heating in winter has been set at 21 °C, which is the operated temperature in the winter [44]. The power density due to lighting and equipment are supposed to be constant during the entire day at 5 W/m$^2$.

*2.4. Scenarios Definition*

The scenarios defined are applied to all buildings of the city of Barcelona, using GIS technology and Cadastral data. Building-by-building methodology is used to apply the scenarios to a total of 750,000 buildings obtaining energy consumption, emissions, energy embodied, and cost implications at an urban scale. In this regard, a typical residential dwelling of Barcelona city (explained in Section 2.2) was studied on 24 scenarios, according to constructive solutions, ventilation, change daily habits, and three proposal energy renovations. Furthermore, two inhabitant behaviours (pre-pandemic and post-pandemic) and the relative position of dwelling with respect to the whole building (mid-floor or top-floor) are considered, according to Section 2.3.

The scenarios studied are defined as showing Table 2 and are explained in the following sections.

**Table 2.** Scenario definition (Source: Own elaboration).

| | Period of Construction | | | Renovation Proposal | | | Dwelling | | Inhabitants Habits | |
|---|---|---|---|---|---|---|---|---|---|---|
| | <1981 | 1981–2007 | >2008 | CR | nZEB1 | nZEB2 | Mid-Floor | Top-Floor | Pre. | Post. |
| 81_MF_Pre | x | | | | | | x | | x | |
| 8107_MF_Pre | | x | | | | | x | | x | |
| 08_MF_Pre | | | x | | | | x | | x | |
| 81_TF_Pre | x | | | | | | | x | x | |
| 8107_TF_Pre | | x | | | | | | x | x | |
| 08_TF_Pre | | | x | | | | | x | x | |
| CR_MF_Pre | | | | x | | | x | | x | |
| CR_TF_Pre | | | | x | | | | x | x | |
| nZEB1_MF_Pre | | | | | x | | x | | x | |
| nZEB1_TF_Pre | | | | | x | | | x | x | |
| nZEB2_MF_Pre | | | | | | x | x | | x | |
| nZEB2_TF_Pre | | | | | | x | | x | x | |
| 81_MF_Post | x | | | | | | x | | | x |
| 8107_MF_Post | | x | | | | | x | | | x |
| 08_MF_Post | | | x | | | | x | | | x |
| 81_TF_Post | x | | | | | | | x | | x |
| 8107_TF_Post | | x | | | | | | x | | x |
| 08_TF_Post | | | x | | | | | x | | x |
| CR_MF_Post | | | | x | | | x | | | x |
| CR_TF_Post | | | | x | | | | x | | x |
| nZEB1_MF_Post | | | | | x | | x | | | x |
| nZEB1_TF_Post | | | | | x | | | x | | x |
| nZEB2_MF_Post | | | | | | x | x | | | x |
| nZEB2_TF_Post | | | | | | x | | x | | x |

Note: MF: mid-floor. TF: top-floor. CR: Conventional Renovation. nZEB1: nZEB renovation type 1. nZEB2: nZEB renovation type 2. Pre: pre-pandemic. Post: post-pandemic.

2.4.1. Renovation Proposals

Three states of renovation are defined. The first one is according to Spanish current standards for conventional building renovation [39]. The second and third one are more exigent strategies based on the recommendations to achieve the European goals for nZEB [45]. Two nZEB states of renovation have been proposed to compare the influence of the envelope insulation level and mechanical ventilation. The three states of renovation are defined as follows (see data in Table 3):

- CR: Conventional energy renovation based on the Spanish current requirements [39].
- nZEB1 renovation: the envelope U-values for façade and roof are those suggested in Spanish regulation to reach the nZEB level of performance [39], windows have a U reduced value and air infiltrations have been also improved to 1 h$^{-1}$ ($n_{50}$). Those values are required by Passivhauss standard for retrofitting (Enerphit standard) [46] for warm climates (Barcelona climate). The airflow ventilation remains similar to CR because it is not a special requirement in the Spanish technical code.
- nZEB2 renovation: using U-value for façade and roof are those required to CR in Spanish regulation [39], which are similar to those required for Passivhauss standard for retrofitting in warm climates. Windows' thermal performance and air infiltra-

tion are as those defined for nZEB1. However, a mechanical ventilation system is included with a Heat Recovery Ventilator (HRV) based on Passivhauss standard for retrofitting [46]. The HRV has an efficiency of 0.75.

**Table 3.** Model parameters for heating demand simulations.

| | | | Unit per m$^2$ | | |
|---|---|---|---|---|---|
| | | **Period** | **Until 1981** | **1981–2007** | **From 2008** |
| Constructive solutions | Façade | U estimated (W/m$^2$ K) | 3.00 | 1.80 | 0.73 |
| | Roof | U estimated (W/m$^2$ K) | 2.50 | 1.40 | 0.41 |
| | Window | U estimated (W/m$^2$ K) | 5.08 | 5.08 | 3.1 |
| Ventilation | | Air infiltration $n_{50}$ (h$^{-1}$) | 9.73 | 5.72 | 4.91 |
| | Ventilation | Type | Open windows | Open windows | OCV (24 h) |
| | | Air flow | 4 h$^{-1}$ during 1 h | 4 h$^{-1}$ during 1 h | 48 L/s |
| Thermal insulation material | Façade | Type | EPS | EPS | EPS |
| | | TC (λ) (W/m K) | 0.035 | 0.035 | 0.035 |
| | | Density (kg/m$^3$) | 35.00 | 35.00 | 35.00 |
| | Roof | Type | XPS | XPS | XPS |
| | | TC (λ) (W/m K) | 0.032 | 0.032 | 0.032 |
| | | Density (kg/m$^3$) | 20.00 | 20.00 | 20.00 |
| Conventional Renovation | Façade | U retrofit (W/m$^2$ K) | 0.49 | 0.49 | 0.49 |
| | | Thickness (m) | 0.06 | 0.05 | 0.02 |
| | | Weight (Kg) | 2.03 | 1.76 | 0.76 |
| | Roof | U retrofit (W/m$^2$ K) | 0.40 | 0.40 | 0.40 |
| | | Thickness (m) | 0.06 | 0.05 | - |
| | | Weight (Kg) | 1.30 | 1.10 | - |
| | Window | U retrofit (W/m$^2$ K) | 2.10 | 2.10 | 2.10 |
| | | Air infiltration $n_{50}$ (h$^{-1}$) | 2.50 | 2.50 | 2.50 |
| | Ventilation | Type | Openings: continuous ventilation (24 h) | | |
| | | Air flow (L/s) | 33.00 | 33.00 | 33.00 |
| nZEB1 renovation | Façade | U retrofit (W/m$^2$ K) | 0.29 | 0.29 | 0.29 |
| | | Thickness insulation (m) | 0.11 | 0.10 | 0.07 |
| | | Weight (Kg) | 3.75 | 3.48 | 2.48 |
| | Roof | U retrofit (W/m$^2$ K) | 0.23 | 0.23 | 0.23 |
| | | Thickness insulation (m) | 0.12 | 0.11 | 0.06 |
| | | Weight (Kg) | 2.48 | 2.28 | 1.17 |
| | Window | U retrofit (W/m$^2$ K) | 1.25 | 1.25 | 1.25 |
| | | Air infiltration $n_{50}$ (h$^{-1}$) | 1.00 | 1.00 | 1.00 |
| | Ventilation | Type | Mechanical ventilation with energy recovery (24 h) | | |
| | | Air flow (L/s) | 33.33. Heat exchanger efficiency: 0.75 | | |
| nZEB2 renovation | Façade | U retrofit (W/m$^2$ K) | 0.49 | 0.49 | 0.49 |
| | | Thickness insulation (m) | 0.06 | 0.05 | 0.02 |
| | | Weight (Kg) | 3.75 | 3.48 | 2.48 |
| | Roof | U retrofit (W/m$^2$ K) | 0.40 | 0.40 | 0.40 |
| | | Thickness insulation (m) | 0.06 | 0.05 | - |
| | | Weight (Kg) | 2.48 | 2.28 | 1.17 |
| | Window | U retrofit (W/m$^2$ K) | 1.25 | 1.25 | 1.25 |
| | | Air infiltration $n_{50}$ (h$^{-1}$) | 1.00 | 1.00 | 1.00 |
| | Ventilation | Type | Mechanical ventilation with energy recovery (24 h) | | |
| | | Air flow (L/s) | 33.33. Heat exchanger efficiency: 0.75 | | |

Note: OCV: Openings of continuous ventilation (24 h). TC: Thermal conductivity.

### 2.4.2. Constructive Solution of Existing Buildings

The constructive solutions of buildings are related to the year of construction. For this study, the typical solutions and transmittances (U-value in W/m$^2$ K) of façade, roof, and window are used according to Spanish regulations for the Barcelona climatic zone (zone C) [32,33]. Table 3 shows the U-value of the envelope elements considered in existing dwellings of building stock and for three renovation states (Section 2.4.1).

### 2.4.3. Air Ventilation and Air Infiltrations

There is no requirement in the Spanish regulations prior to 2007 referred to air ventilation in dwellings. Therefore, for these buildings, the air flow ventilation is carried out by opening windows. For dwellings built after 2008, ventilation flows are those that appear in the Spanish regulations in force in each period of time [47].

The mechanical air ventilation system including heat recovery is decisive for reducing the air ventilation losses and, consequently, is an obligatory requirement for the Passivhaus standard. The minimum acceptable heat exchanger sensible efficiency is 75%, which is a typical value for heat exchangers currently available on the market. Therefore, for the nZEB2 renovation scenario, mechanical ventilation is considered with the required air flow for four people occupancy (the nominal ventilation flow recommended by Passivhaus is 30 $m^3$/h per person).

The air change rate $n_{50}$ ($h^{-1}$) due to air infiltrations throughout the building envelope depends on many factors. The building's year of construction has been identified in recent literature as one important parameter to estimate the level of air infiltrations. Furthermore, the $n_{50}$ values in Table 3 depending on the year of construction have been found for Barcelona city [48,49]. Spanish regulations neither require nor recommend any maximum value for air infiltrations, which are actually important when increasing the quality of the building enclosures. For this reason, for CR and nZEB1, an air renovation rate of 2.5 $h^{-1}$ have been selected for heating demand simulations. The $n_{50}$ value for nZEB2 renovation comes for the Passivhaus standard for retrofitting [46].

The degree of infiltration included in the model (in real conditions and real pressure drop) is calculated from the estimated value of $n_{50}$ as follows [50].

$$N_{average} = n_{50}/N \qquad (1)$$

where N is a correlation factor depending mainly on the climatic conditions and the shielding of the building. The value of N = 21.81 obtained from Guillén-Lambea et al. [50] for Barcelona city has been used for all scenarios.

### 2.5. Energy Consumption and GHG Emissions

The heating energy demand has been estimated using TRNSYS software [51] for the 24 defined scenarios in Table 2. The TRNSYS dwelling model allows obtaining the energy demand to maintain the interior conditions required in the dwelling (temperature and relative humidity) under particular conditions (model inputs) such as climate data, occupancy, internal loads, and air infiltrations. The heat balance is done in discrete time, usually for one year in one-hour intervals. A simulation model has been developed for each of the two dwellings selected (mid-floor and top floor). For each construction period and for the three renovation proposals, then the model has been run with the two conditions related to the daily habits of the inhabitants.

The climatic data have been included in the model for the city of Barcelona. The SWEC (Spanish Weather for Energy Calculations) file has been used from the EnergyPlus weather database [52] and the simulations have been carried out on an hourly basis for one year.

The heating energy demand has been obtained for the 24 scenarios and the heating energy consumption has been calculated by applying the breakdown of heating consumption by energy sources published by IDAE (2012) [53] for the Spanish Mediterranean area where 66.8% is from electricity, 20.4% from natural gas, 4.0% from liquefied petroleum gas (LPG) coal, 5.7% from oil, 0.6% from coal, and 2.5% from renewables. Consistent performances have been estimated based on the maturity of the technologies and on Thermal Installations Performance Document for Heating Equipment [54,55]. The conversion factors used were 0.9 for natural gas and oil boilers, 1 for electric radiators, and 0 for renewable energies.

Table 4 shows the energy transition factors from final energy consumption (kWh) to emissions (in kg of $CO_2$) from different sources of energy consumed in the building sector in Spain (Joint Resolution of the Ministries of Industry, Energy and Tourism and Ministry of Development), published in 2017 [56].

**Table 4.** Transition factors to $CO_2$ emissions.

| Fuel | Kg $CO_2$/kWh |
|---|---|
| Electricity (peninsular) | 0.331 |
| Heating oil | 0.311 |
| LPG | 0.254 |
| Natural gas | 0.252 |
| Coal | 0.472 |

*2.6. Life Cycle Assessment*

The environmental implications of conventional and nZEB renovations has been calculated by means of the Life Cycle Assessment (LCA) methodology [57]. This study follows a cradle-to-site approach, including the production of building materials, their transportation to the building site, and their installation [58]. Table 3 summarises the thickness of insulation material installed in each proposal for façade and roof, the requirement of windows, and, in the case of the nZEB2 renovation, the installation of a ventilation system. The functional unit (FU) selected for this study, according to the environmental product declaration (EDP) for construction products EN 15804:2012+A2, 2020, was 1 m$^2$ of the different solutions [59,60]. Subsequently, the FU was applied to the total area of each constructive solution in each building.

The environmental information for the characterization of related impacts has been gathered from the Ecoinvent 3.6 [61] database of SimaPro version 9 [62]. The impact category included in this study is the Global Warming Potential (GWP) due to its relevance in the environmental field, and the Cumulative Energy Demand or Embodied Energy (EE) for the subsequent comparison with operating energy of buildings.

*2.7. Economic Cost*

This study estimates the execution costs for the buildings of each construction period to achieve the measures of the three improvement scenarios, by means of the CYPE software and database [63], used in other studies [40].

Table 5 shows the enforcement costs to retrofit a typical building to achieve conventional renovation, nZEB1, and nZEB2 renovation levels. This cost depends on the insulation added to the existing elements, according to Table 3. They are shown per m$^2$ of constructive solution in the case of façade, roof, and window, and per m$^2$ of dwelling in the case of mechanical ventilation and tightness.

**Table 5.** Enforcement costs of renovation.

| Title 1 | Constructive Element | Conventional Renovation (€/m$^2$) | nZEB1 Renovation (€/m$^2$) | nZEB2 Renovation (€/m$^2$) |
|---|---|---|---|---|
| Built before 1981 | Façade | 98.4 | 112.7 | 98.4 |
| | Roof | 46.7 | 62.7 | 46.7 |
| | Window | 359.4 | 398.6 | 398.6 |
| | Mechanical Ventilation, tightness | - | - | 113.2 |
| Built between 1981–2007 | Façade | 96.4 | 108.6 | 96.4 |
| | Roof | 44.7 | 56.8 | 44.7 |
| | Window | 359.4 | 398.6 | 398.6 |
| | Mechanical Ventilation, tightness | - | - | 113.2 |
| Built after 2008 | Façade | 92.4 | 100.5 | 92.4 |
| | Roof | 0 | 46.7 | 0 |
| | Window | 359.4 | 398.6 | 398.6 |
| | Mechanical Ventilation, tightness | - | - | 113.2 |

*2.8. Extrapolation from the Building Level to the Urban Level*

Once the calculations of energy consumption, environmental impact, and cost by m$^2$ is done, they are applied to each building of Barcelona, which, in total, includes 60,000 buildings. The method to determine the simplified net building living area (coverage × height

index) follows the bottom-up methodology from García-Pérez et al. [28]. This method characterizes quantitatively the envelope surface by developing a geospatial model using the QGIS software [64], and the cadastral vector cartography information from the Spanish Government [65]. As the Cadastral dataset includes data about the use of buildings, it is possible to identify for each Cadastral reference (plot) of the total area from each residential building with heating needs. In this study, only the physical characteristics of the building obtained from Cadastre that affect the thermal renovation and the calculation of the heating energy consumption of each residential building have been included. The physical characteristics that allow us to calculate the methodology used are: total coverage (including façade and roof area), gross living area (including all space that has heating, lighting, and ventilation), and height of living index (being the relation between the coverage and the gross living area).

Based on the calculated physical characteristics of each building, several floors are assigned from the height index and considered 3 m as the average height of each floor [32]. All buildings have a top-floor, with different consumption patterns from mid-floors, but with the same coverage (Figure 2). Based on that, the energy consumption of each building is determined. With respect to LCA and cost for each renovation, it is calculated using the heated envelope, differentiated by façade, roof, and windows. The ventilation system is installed for the nZEB2 renovation proposal.

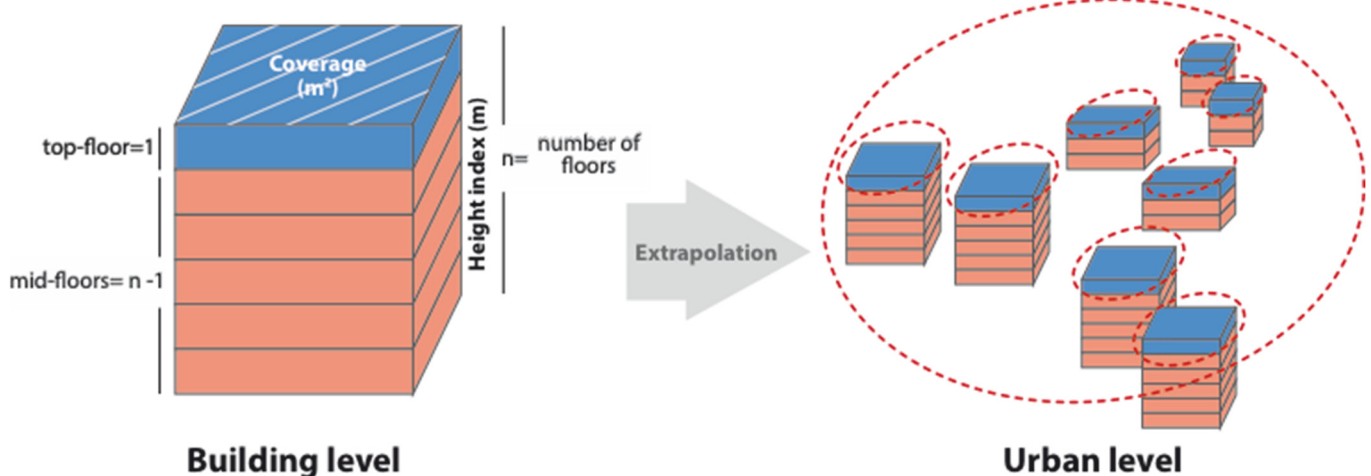

**Figure 2.** Schematic diagram of the process of extrapolation.

After the building is characterized, it is possible to extrapolate results at the urban level and determine the energy consumption, environmental impact, and economic implications at the urban scale. The results are presented directly and through scenarios to fully understand the scope of these impacts.

## 3. Results and Discussion

### 3.1. Case Study: Barcelona (Spain)

In Figure 3, buildings have been coloured in different tones according to the groups defined by age of construction and the data is shown in Table 6. The data obtained reveal the aging of residential buildings in the city of Barcelona. As such, 87.6% of residential buildings were built before 1981, 11.1% were built between 1981 and 2007, and only 1.3% were recently built, in the line of other studies [66]. The results change slightly when the percentages of dwellings are analysed since the buildings built between 1981 and 2007 are blocks with a greater number of dwellings than those built in previous years and, therefore, the percentage increases slightly to 13.8%. Dwellings built before 1981 represent 84.3% of the city's building stock and those built after 2007 represent only 1.9% of all city buildings.

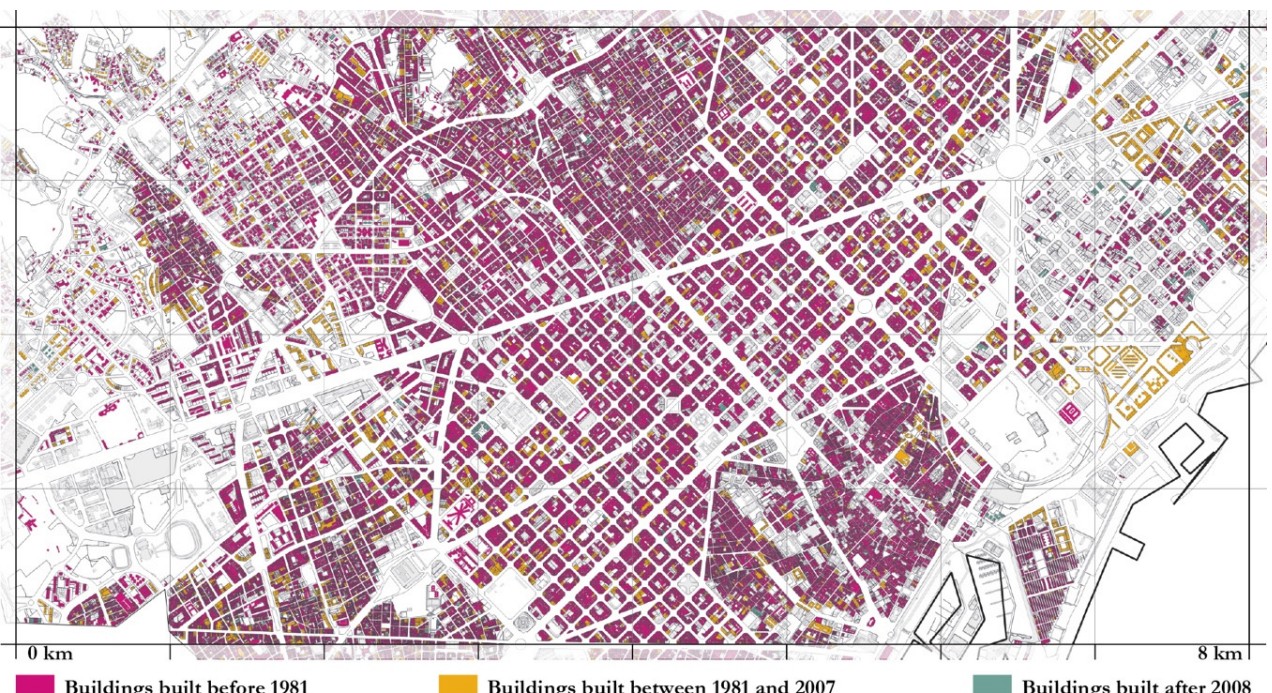

**Figure 3.** Age of residential buildings by the construction period in Barcelona (Spain).

**Table 6.** General information of the case study: Barcelona (Spain) (data obtained from the Cadastre source).

| | Total Number of Residential Buildings | Total Number of Dwellings | Total Heated Surface (m$^2$) | Total Thermal Envelope to Retrofit (m$^2$) |
|---|---|---|---|---|
| Built before 1981 | 53,854 | 691,416 | 60,594,426 | 143,666,702.46 |
| Built between 1981–2007 | 6807 | 112,899 | 11,342,004 | 34,099,289.60 |
| Built after 2008 | 820 | 15,788 | 1,310,359 | 5,306,380.21 |

The percentage of residential buildings built before 1981 (87.6%) is higher in Barcelona than the average of Spain (56.3%). This is because Barcelona had a construction explosion in the 1970s. Furthermore, its physical limits prevent the city from growing, which is why most of the new buildings are built in metropolitan municipalities. These data demonstrate the need for renovation of the city's building park in order to meet the energy efficiency requirements of cities.

The methodology developed in this paper, using Cadastral and GIS data, allows us to obtain information about the Barcelona city, as shown in Table 6. The 83% of a residential heated surface in the city of Barcelona correspond to dwellings built before 1981 and only 2% corresponded to dwellings built after 2008. Furthermore, the city of Barcelona has 183,072,372 m$^2$ of thermal envelope (façade, roof, and windows) to retrofit. This data is used to determine the total impacts at city scale.

### 3.2. Results: LCA and Heating Demand Simulation

Table 7 shows the heating energy consumption and LCA of the scenarios defined per m$^2$. The change in daily habits due to the pandemic can increase energy consumption and $CO_2$ emissions by 182% in the case of no retrofit. Considering the post-COVID-19 situation, conventional renovation achieves between 42% and 81% energy and emissions saving, nZEB1 achieve between 75% and 91% of savings and nZEB2 between 92% and 97%. Once U-values of the envelope required in conventional renovation is achieved, the better way to achieve energy savings is installing a mechanical ventilation system with energy recovery instead of adding more insulation (considering the heating system constant). The results confirm the recommendation found on the recent literature to include

heat recovery in ventilation systems to minimize the demand for heating in homes even in mild climates to achieve minimum levels of energy consumption [37]. Furthermore, the ventilation system is the less impacted element of retrofit according to LCA (per m$^2$). Windows are the retrofit element with the most embodied energy, but the substitution has additional benefits, such as reducing the air infiltration and acoustic insulation [39].

**Table 7.** Heating consumption and LCA results by year of construction and type of renovation depending on inhabitant habits.

| | Scenario | Year of Construction | Heating Consumption (kWh/m$^2$·y) | CO$_2$ Emission by Heating (kg CO$_2$-eq/m$^2$·y) | Retrofit Façade (kg CO$_2$-eq/m$^2$) | Retrofit Roof (kg CO$_2$-eq/m$^2$) | Retrofit Windows (kg CO$_2$-eq/m$^2$) | Retrofit Ventilation System (kg CO$_2$-eq/m$^2$) |
|---|---|---|---|---|---|---|---|---|
| **PreCovid-19** | No retrofit | <1981 | 117.18 | 31.24 | - | - | - | - |
| | | 1981–2007 | 72.50 | 19.33 | - | - | - | - |
| | | >2008 | 42.83 | 11.41 | - | - | - | - |
| | Conventional renovation | <1981 | 24.56 | 6.54 | 28.00 | 11.57 | 289.65 | - |
| | | 1981–2007 | 24.56 | 6.54 | 27.08 | 10.81 | 289.65 | - |
| | | >2008 | 24.56 | 6.54 | 23.69 | 0 | 289.65 | - |
| | nZEB1 renovation | <1981 | 9.32 | 2.49 | 33.85 | 16.03 | 303.29 | - |
| | | 1981–2007 | 9.32 | 2.49 | 32.92 | 15.27 | 303.29 | - |
| | | >2008 | 9.32 | 2.49 | 29.54 | 11.11 | 303.29 | - |
| | nZEB2 renovation | <1981 | 2.44 | 0.65 | 28.00 | 11.57 | 303.29 | 9.93 |
| | | 1981–2007 | 2.44 | 0.65 | 27.08 | 10.81 | 303.29 | 9.93 |
| | | >2008 | 2.44 | 0.65 | 23.69 | 0 | 303.29 | 9.93 |
| **PosCovid-19** | No retrofit | <1981 | 209.86 | 55.94 | - | - | - | - |
| | | 1981–2007 | 132.20 | 35.24 | - | - | - | - |
| | | >2008 | 70.56 | 18.81 | - | - | - | - |
| | Conventional renovation | <1981 | 40.77 | 10.88 | 28.00 | 11.57 | 289.65 | - |
| | | 1981–2007 | 40.77 | 10.88 | 27.08 | 10.81 | 289.65 | - |
| | | >2008 | 40.77 | 10.88 | 23.69 | 0 | 289.65 | - |
| | nZEB1 renovation | <1981 | 17.91 | 4.78 | 33.85 | 16.03 | 303.29 | - |
| | | 1981–2007 | 17.91 | 4.78 | 32.92 | 15.27 | 303.29 | - |
| | | >2008 | 17.91 | 4.78 | 29.54 | 11.11 | 303.29 | - |
| | nZEB2 renovation | <1981 | 5.96 | 1.59 | 28.00 | 11.57 | 303.29 | 9.93 |
| | | 1981–2007 | 5.96 | 1.59 | 27.08 | 10.81 | 303.29 | 9.93 |
| | | >2008 | 5.96 | 1.59 | 23.69 | 0 | 303.29 | 9.93 |

Table 8 shows the global heating energy consumption, environmental implications, and cost results at an urban scale applied to the city of Barcelona. The data presented as the total impact for retrofit in kgCO$_{2eq}$ are the emissions due to the fabrication and the renovation works to implement the needed new materials (envelope insulation, window . . . ) and the impacts savings in kgCO$_{2eq}$ are the emissions that would be avoided if the existing buildings were renovated under the conditions of the three proposed renovation scenarios during the building use for a lifetime of 30 years. Moreover, the embodied energy needed to perform the building renovations are compared to the energy impacts savings in case of building retrofitting, considering the change in daily habits by pandemic and maintaining a renovation lifetime of 30 years (until 2050).

Results show that, the older the building is, more kgCO$_{2eq}$ and energy savings are achieved with the retrofit. Retrofitting all buildings of Barcelona built before 1981, could save between $1.5 \times 10^8$ tons and $1.7 \times 10^8$ tons of CO$_2$ in 30 years. For <1981 period, nZEB renovations only achieve 13% and 15% more of kgCO$_{2eq}$ savings in comparison with CR. Furthermore, CR has the shorter payback time, nZEB2 renovation has slightly lower embodied energy than nZEB1. However, nZEB1 has a shorter payback time.

Retrofitting buildings built between 1981 and 2007 implies a reduction on kgCO$_{2eq}$ emission and an energy saving of 10% with respect to <1981 period. Furthermore, the payback time is longer, between 179% and 207%. In this case, nZEB renovations compared to the conventional, achieve 20% (nZEB1) and 27% (nZEB2) for both kgCO$_2$-eq avoided and energy savings. CR and nZEB1 have similar payback time, and the nZEB2 has the higher. However, the nZEB1 renovation has higher environmental implications than nZEB2 and CR.

In the case of retrofitting buildings built after 2008, the reduction on $kgCO_{2eq}$ emitted and energy savings are less than 1% with respect to buildings built before 1981. nZEB renovations achieve $kgCO_{2eq}$ emissions saving between 150% (nZEB1) and 162% (nZEB2) with respect to CR, but the payback time are too long, and are close to the lifetime.

In general terms, CR has shorter payback time regardless of the construction period of the building, but the $kgCO_{2eq}$ avoided and energy saving are the lowest. Retrofitting all buildings of Barcelona according to CR instead of nZEB will produce $2.25 \times 10^7$ tons of $CO_2$ in the case of nZEB1 and $2.57 \times 10^7$ tons of $CO_2$ in the case of nZEB2 during the 30 years of use of the building park. Comparing the two nZEB renovations, proposals for the three periods, nZEB2 has the best results: the lower embodied energy and the higher reduction for both $kgCO_{2eq}$ and energy consumption. However, the payback results are similar in the buildings built before 1981 and between 1981 and 2007, but it is contrary more recently. In the case of the first and second period of construction, nZEB2 renovation has 40% longer payback, but only 1% and 6% more impact savings. In the case of buildings erected after 2008, nZEB2 achieve 5% more impact savings and has four years less of payback.

Table 9 shows the balance between impact savings for retrofit in post-pandemic scenarios for $CO_2$ emissions, energy, and costs, per $m^2$ of the heated surface. In environmental impacts and energy saving terms, the most advisable option for retrofit for all periods would be nZEB2. In economic terms, the most advantageous option is nZEB1, except for the most recent buildings built.

The $CO_2$ emissions reduction by $m^2$ of the buildings surface when retrofitting the older building to a CR is 1.92 times higher than when retrofitting a building from 1981–2007 and 11.7 times higher when retrofitting new buildings. This ratio is 1.8 and 5.3 times higher when retrofitting to nZEB1 and 1.7 and 4.6 times higher when retrofitting to nZEB2. These results show the importance and the relevance of renovating the oldest buildings in a city from an energy efficiency and emission reduction point of view. From an economic point of view, the data shows that renovating older buildings to the conditions defined for nZEB1 is the most advantageous option.

**Table 8.** Total results at an urban level depending on the renovation type for a post-pandemic situation, city of Barcelona.

| | | Total Impact for Retrofit (Mt $CO_2$-eq *) | Impact Saving for Retrofit in Post-COVID-19 for 30 Years (Mt $CO_2$-eq *) | Total Embodied Energy Retrofit (TJ **) | Total Energy Saving for Retrofit in Post-COVID-19 for 30 Years (TJ **) | Total Economic Invest for Retrofit (Billion € ***) | Total Economic Saving for Retrofit in Post-COVID-19 for 30 Years (Billion € ***) | Average Payback for Retrofit in Post-COVID-19 per m$^2$ (Years) |
|---|---|---|---|---|---|---|---|---|
| | CR | 3.84 | 153.04 | $8.6 \times 10^4$ | $2.1 \times 10^6$ | 10 | 64.26 | 4.82 |
| <1981 | nZEB1 | 4.58 | 169.52 | $1.0 \times 10^5$ | $2.3 \times 10^6$ | 12.41 | 71.18 | 5.3 |
| | nZEB2 | 4.53 | 172.08 | $9.6 \times 10^4$ | $2.3 \times 10^6$ | 17.1 | 72.25 | 7.6 |
| | CR | 0.77 | 14.89 | $1.7 \times 10^4$ | $2.0 \times 10^5$ | 2.13 | 6.25 | 10.0 |
| 1981–2007 | nZEB1 | 0.93 | 18.32 | $2.2 \times 10^4$ | $2.5 \times 10^5$ | 2.57 | 7.70 | 9.7 |
| | nZEB2 | 0.89 | 18.59 | $1.9 \times 10^4$ | $2.5 \times 10^5$ | 3.45 | 7.80 | 13.6 |
| | CR | 0.05 | 0.32 | $9.1 \times 10^2$ | $4.3 \times 10^3$ | 0.10 | 0.13 | 21.7 |
| >2008 | nZEB1 | 0.11 | 0.78 | $2.6 \times 10^3$ | $1.1 \times 10^4$ | 0.32 | 0.33 | 27.7 |
| | nZEB2 | 0.07 | 0.77 | $1.1 \times 10^3$ | $1.0 \times 10^4$ | 0.25 | 0.32 | 23.7 |

Note: CR: conventional renovation. nZEB1: nZEB renovation case 1. nZEB2: nZEB renovation case 2. * 1 Mt (megatons) = 106 t. ** 1 TJ = 106 MJ. *** 1 Billion € = 1000 million € = 109 €.

**Table 9.** Net total saving and in relation to a heated surface.

| | | Net Impact Saved for Retrofit (Mt $CO_2$-eq *) | Net Impact Saved for Retrofit per Heating Surface (Ton $CO_2$-eq/m$^2$) | Net Energy Saved for Retrofit (TJ **) | Net Energy Saved for Retrofit per Heating Surface (MJ/m$^2$) | Net Cost Saved for Retrofit (Billion € ***) | Net Cost Saved for Retrofit per Heating Surface (€/m$^2$) |
|---|---|---|---|---|---|---|---|
| | CR | 149.2 | 2.41 | $2.01 \times 10^6$ | 33,237.38 | 54.26 | 891.17 |
| <1981 | nZEB1 | 164.94 | 2.72 | $2.20 \times 10^6$ | 36,306.97 | 58.77 | 973.69 |
| | nZEB2 | 167.55 | 2.76 | $2.20 \times 10^6$ | 36,372.98 | 55.15 | 910.97 |
| | CR | 14.12 | 1.25 | $1.83 \times 10^5$ | 16,134.71 | 4.12 | 361.49 |
| 1981–2007 | nZEB1 | 17.39 | 1.50 | $2.28 \times 10^5$ | 20,102.27 | 5.13 | 449.66 |
| | nZEB2 | 17.7 | 1.59 | $2.31 \times 10^5$ | 20,366.77 | 4.13 | 379.12 |
| | CR | 0.27 | 0.21 | $3.39 \times 10^3$ | 2587.08 | 0.03 | 25.95 |
| >2008 | nZEB1 | 0.67 | 0.51 | $8.40 \times 10^3$ | 6410.46 | 0.01 | 7.63 |
| | nZEB2 | 0.7 | 0.60 | $9.30 \times 10^3$ | 7097.29 | 0.07 | 53.42 |

* 1 Mt (megatons) = $10^6$ t. ** 1 TJ = $10^6$ MJ. *** 1 Billion € = 1000 million € = $10^9$ €.

## 4. Limitations of This Study and Directions for Further Analyses

This section presents the methodological limitations of this study. When working at an urban level using a bottom-up methodology for an extrapolation based on data at a building level, there are different aspects to consider. The methodology used is universal. However, local aspects and local data should be used for calculations in other locations. To apply this methodology to another location, the following data are necessary: cadastral data, local residential buildings, and dwelling characteristics (height, surface area, constructive solutions, window area . . . ), climate conditions, local building codes, and their change in the past, costs, and LCA data of the region. The methodology on which this study is based already indicated that these aspects have a direct relationship in the results obtained but proportionate to all buildings analysed. Therefore, this methodology is still valid according to its objectives, presenting data at the urban level and being able to identify areas of interest.

The buildings of the city are divided in three periods, and each period has assigned envelope characteristics according to the literature. On some occasions, these characteristics, such as the U-value of the envelope, can be unfavourable. However, they are useful for establishing the potential at an urban scale. Furthermore, to this study, a typical dwelling is considered to obtain the data per m$^2$. Then, this data is extrapolated to Barcelona buildings. Therefore, this study does not consider the orientation and shadows of each Barcelona building because they are aspects that remain constant in all the compared scenarios. On the contrary, from each building, it has calculated the façade and roof area, the total heated area, number of dwellings, etc. For that, physical aspects that are relevant to the purpose of the study have been analysed: energy renovation and energy consumption of residential buildings. To achieve nZEB requirements in Spain, the building must have very low energy consumption due to heating, cooling, and domestic hot water (DHW). In this study, only heating consumption has been considered. To obtain this requirement, this study considers two ways to achieve it: nZEB1 and nZEB2. These two renovation scenarios do not consider the specific characteristics of the dwellings in order to adapt it. For example, in the case of modifying the ventilation of dwellings, it would be necessary to carry out works inside the buildings and reduce height in some areas. In this study, these interventions have been considered feasible in each dwelling. Regarding the post-COVID-19 scenario, the heating simulation assumes that, in all dwellings, at least one person modifies their habits and spends more time at home. This assumption allows us to obtain a potential to increase consumption. However, it is possible that a person cannot telework in all homes.

With respect to energy consumption, only the impact associated with heating energy demand for each proposed scenario has been calculated to keep the influence in cooling energy demand out on the study scope. First, because the heating energy consumption for residential buildings located in the Mediterranean area, as Barcelona city, represents the 40.9% of the total energy consumption in residential buildings while that of cooling is only 1.1% [36]. Second, the reduction expected in cooling energy consumption after a dwelling renovation is not as important as the reduction achieved in heating energy consumption. This fact is not only due to the fact that the heating consumption is higher than that of cooling, but also that the improvements made in the quality of the envelope and the enclosures, ventilation strategies, or the dwelling airtightness enhancements do not help in a considerable way for the reduction of energy consumption for cooling. Likewise, the following aspects such as increased use of energy based on artificial lighting, electric vehicular charges, use of energy efficient lighting such as LEDs, and use of lighting control strategies is out of the study scope and should be considered for future research.

With respect to the LCA, economic cost methodology is based on the same type of thermal retrofit for all buildings, which is, in this case, external thermal insulation. This has been done in this way due to being a type of non-invasive, less expensive, and more effective rehabilitation to avoid thermal bridges. In certain buildings, due to their structure or uniqueness, their external insulation is not possible. Other internal insulation options

should be considered. For that, LCA and economic cost results can be useful to obtain a general image of the situation of rehabilitation of the buildings responding to their thermal needs, according to their year of construction. An additional limitation of the LCA method is the use of generic databases that, in some cases, do not include specific aspects of the environment to be analysed, but that these inaccuracies are constant for all the elements of the study.

Future research derived from this study could include the mentioned aspect out of its scope. This study gives results at some level of approximation, and better numerical results can be obtained if more factors are included into the calculation. In this regard, the consideration of aspects that remain constant in all the compared scenarios, e.g., orientation or shading by adjoining buildings, could refine energy models approaching more realistic consumption patterns, which also include new domestic habits. In addition, considering aspects such as morphology and socio-economic and demographic information in the extrapolation method, could generate more accurate results. Finally, this methodology could include the analysis of other energy consumptions in addition to heating, such as cooling, and being related to other parameters listed above, such as sunny and shaded areas, sunscreen, and orientation of the buildings.

## 5. Conclusions

The COVID-19 disease has changed the inhabitant's daily habits in residential buildings. The more time users spend at home can increase energy consumption and retrofitting the old building stock is becoming more necessary. This paper develops a methodology using Cadastral and GIS data to determine the energy, environmental, and economic cost implications of the retrofit applied to the city of Barcelona (Spain). A total of 24 scenarios were defined according to the year of construction, constructive solutions, inhabitant´s daily habits, and three retrofit proposals. The proposed renovation scenarios are conventional energy renovation, according to Spanish requirements to retrofit, nZEB1 renovation improving the insulation of the buildings envelope, and nZEB2 renovation including a mechanical ventilation system and a heat recovery ventilator.

The proposed methodology allows us to assess the environmental, energy, and economic implications for an unexpected situation, such as COVID-19 crisis, that completely changes the daily routines of a city, a country, a continent, etc. This methodology is based on a flexible aggregated model that works from the building to an urban scale. Therefore, it allows us to obtain relevant physical characteristics of the buildings' stock of a city from cadastral data, and estimate the energy, environmental implications, and cost implications of their renovation. The methodology is universal, and their flexibility in scenarios and renovation definition allows us to apply other European countries, even the study of the effect of changes in life habits not related to the pandemic. The obtained results are geo-referenced by facilitating the understanding of results by stakeholders and policy makers and being possible to identify priority areas to intervene. However, this methodology is not exempts of limitations. Working with energy models of the dwellings, as well as of the building stock model, imply a simplification of the complex urban reality. Future research could deal with the relationship between these energy performance models and real consumption data at an urban scale, in order to test and refine their effectiveness. Furthermore, the extrapolation at an urban scale could consider more urban attributes, such as urban morphology, demography, or other socio-economic/spatial patterns that could help to obtain more fine grain results.

According to the results, the residential buildings built before 1981 in Barcelona city (87.6%) are higher than the average of Spain (56.3%). This can be a generalized fact in cities whose expansion occurred in the 60–70 s and reinforce the need to invest in a retrofit in the cities, where the majority of the existing buildings are old and inefficient in terms of energy. Furthermore, the results of this study show that a change in daily habits due to the pandemic can increase the energy consumption and $CO_2$ emission in residential buildings in 182%.

The environmental impacts due to the retrofit and the impact savings for retrofit in a post-pandemic situation (heating energy consumption and $CO_2$ emissions) are compared. The results show that the $CO_2$ emissions reduction by m$^2$ of the buildings' surface when retrofitting the older building to a CR is 1.92 times higher than when retrofitting a building from 1981–2007, and 11.7 times higher when retrofitting new buildings. This ratio is 1.8 and 5.3 times higher when retrofitting to nZEB1 and 1.7 and 4.6 times higher when retrofitting to nZEB2. Furthermore, retrofitting buildings built before 1981 achieves net energy savings until 2.0 times higher than 1981–2007 and 12.8 times higher than buildings erected after 2008. Therefore, we concluded that the older buildings achieve more savings in the balance, so the buildings erected before 1981 obtained better energy savings after the renovation. Retrofitting buildings built in this period has a lower economic payback time. However, when retrofitting the buildings erected after 2008, although the energy and emissions balance is positive, the payback time is between 21 and 28 years.

The nZEB2 renovation (with energy recovery) is the option with better energy and emission savings, but also is an option with higher payback time for buildings built until 2007. Retrofitting all buildings of Barcelona built before 1981, could save between $1.5 \times 10^8$ tons and $1.7 \times 10^8$ tons of $CO_2$. Retrofitting all buildings of Barcelona according to CR instead of nZEB will produce $2.25 \times 10^7$ tons of $CO_2$ in the case of nZEB1 and $2.57 \times 10^7$ tons of $CO_2$ in the case of nZEB2.

**Author Contributions:** Conceptualization, J.S.-P. Methodology, J.S.-P., M.M.-C., and S.G.-L. Software, J.S.-P., M.M.-C., and S.G.-L. Validation, J.S.-P., M.M.-C., and S.G.-L. Formal analysis, J.S.-P. and S.G.-P. Investigation, J.S.-P., M.M.-C., and S.G.-L. Resources, A.L.M.-G. Data curation, S.G.-P. and A.L.M.-G. Writing—original draft preparation, M.M.-C. Writing—review and editing, J.S.-P., M.M.-C., S.G.-L., and S.G.-P. Visualization, J.S.-P. and S.G.-P. Supervision, J.S.-P. Project administration, J.S.-P. Funding acquisition, J.S.-P. All authors have read and agreed to the published version of the manuscript.

**Funding:** This research was funded by the Universidad de Zaragoza, project number UZ2020-TEC-07.

**Data Availability Statement:** Part of datasets analysed in this study are publicly available. This data can be found here: Spanish Cadastre (https://www.sedecatastro.gob.es/), The prices canbe found in CYPE price generator (http://www.generadordeprecios.info/#gsc.tab=0), Part of used data in this study has restrictions apply to the availability of these data. Data was obtained from Simapro and Trnsys software and are available with the required license of use.

**Conflicts of Interest:** The authors declare no conflict of interest.

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
