# Peer review of "Heating Energy Consumption and Environmental Implications Due to the Change in Daily Habits in Residential Buildings Derived from COVID-19 Crisis: The Case of Barcelona, Spain"

_sustainability, doi:10.3390/su13020918_

Round 1

Reviewer 1 Report

I thank the authors for this interesting research submission.

General comments:
Introduction

The introduction should be extended and deepened by a critical-comparative method that provides a complete and exhaustive state of the art analysis.

As this paper was recently submitted, I wonder why the authors have not checked to see if similar papers connected to energy implications and covid-19 have been performed.

A simple google search has indicated the following papers, including one from Spain, which should be added to the Introduction. (This particular paper presents a detailed analysis of how confinement measures have modified electricity consumption in Spain during the covid-19 lockdown.)

https://doi.org/10.1016/j.enpol.2020.111964

https://doi.org/10.1016/j.isci.2020.101639

https://doi.org/10.1002/er.6007

https://doi.org/10.1016/j.heliyon.2020.e05202

https://doi.org/10.3390/en13133357

https://doi.org/10.1525/curh.2020.119.820.317

https://doi.org/10.1080/15567036.2020.1801902

https://doi.org/10.1016/j.erss.2020.101654

Line:28
“Retrofitting all buildings of Barcelona according to conventional energy renovation instead of nZEB will  produce between 2,25*107 and 2,57*107 tons of CO2.“

Don’t use abbreviations in the abstract.

Line:34
I would like to see a diagram indicating the amount of daylight hours available in Barcelona, where solar energy was utilized in 2020.

Line:40
Some part of the text seems to be missing. I suggest you use reference [3] and incorporate it at the end of the sentence for improved readability.

Line:112
“The research methodology includes the next steps developed in the following sections. The  different steps of the methodology are summarized in Figure 1.”

Please rewrite this text. Why has lighting been taken out of this equation?

Line:162
Please add diagrams for these three studies here.

Line:179
“For lighting and equipment, a nominal load of 5 W/m2 has been  considered.”

Are you referring to “lighting power density”?

5W/m2 will vary based on light sources, it seems to be rather low.

What relevant energy classes are we talking about?

The 3.2 EU GPP Criteria for designing indoor lightning for residential buildings allows 11w/m2 and for cellular office – 13.

Line:182
(Table 1), L196 (Table 2), 340 (Table 6). Please provide your source of reference.

Line:423-429
Limitations and further works need to be listed as a separate chapter.

Are there any other limitations?

The following aspects such as increased use of energy based on artificial lighting, electric vehicular charges, use of energy efficient lighting such as LEDs, use of lighting control strategies to class A, should be considered for future research.

Line:438
“The results show that the older buildings achieve more savings in the balance, so the buildings erected before 1981 obtained the better energy savings after the renovation.”

What is the reason, please indicate it.

Line:448-457 
I would suggest only using initials instead.

Line:468-469
Please add more information.

Reviewer 2 Report

The article deals with the impact of changes in the behavior of users of residential buildings caused by the COVID-19 crisis on energy consumption and the resulting increase in CO2 emissions. This issue is very important and extremely topical nowadays.

Unfortunately, the publication does not adequately define the "post-covid" status - what percentage of the population and to what extent will be affected by the change in the behavior of building use in the long term (i. e. following the widespread introduction of vaccination against SARS-CoV-2). The model used by the authors is based on the assumptions of a constant - higher - temperature in apartments and a constant level of internal thermal loads. These assumptions are applied to all apartments in the city. Such an approach seems to correspond rather to the lockdown of the whole economy, and may only apply for the next few months. For this reason, the results for the entire life cycle of buildings will be overestimated.

Then, the article discusses only the changes in heat consumption for heating during the winter period. Meanwhile, both changes in the habits of apartment users and the modernisation of buildings have an impact on energy consumption throughout the year, including energy consumption for cooling in summer, especially in warm climates.

Furthermore, it seems that adopting a detailed analysis of two dwellings as a basis for modelling is an oversimplification that does not take into account all relevant building parameters such as a envelope area/volume factor orientation towards the world's sides, influence of neighbouring buildings, etc.

Additionally, the description of the transition from building to city level is too general. In particular, no information was given as to whether the modelling of the thermomodernization process, especially to the nZEB level, took into account the lack of possibility to perform certain activities in historic buildings, which are numerous in Barcelona.

In conclusion, the methodology used yields results that are too error-prone to draw constructive conclusions at the level of the entire city, especially in the life cycle of buildings.

It will be difficult to generalise the results or apply this methodology to other European countries, let alone the global scale, while the publication should have a universal dimension.

Round 2

Reviewer 2 Report

Dear Sirs,

Thank you very much for sending your reply to my review of the manuscript “Heating energy consumption and environmental implications due to the change in daily habits in residential buildings derived from COVID-19 crisis. The case of Barcelona, Spain.” and for the comprehensive responses to my comments you have given.

I suggest that you add the following information to the Abstract, in order to highlight the universal aspects of your study: The methodology presented can be applied in any city with sufficient cadastral data, and is considered optimal in the European context as it goes for calculating the heating energy consumption.

Also, I would state explicitly in the Introduction that the study, and all its implications at the city scale and LCA calculations, refer to the existing residential building stock in Barcelona (i.e. new housing and demolitions have not been included in the study).

Your explanations address almost all the issues under consideration. Unfortunately, it still remains unclear for me why the differences between energy use for cooling and heating in Barcelona are so high. Please specify what type of cooling do you refer to – for instance, what about energy for cooling used for split air conditioners that people install in their appartements?

Then, I suggest that chapter 4 is called “Limitations of this study and directions for further analyses.” and partly rewritten. Namely, the study gives results at some level of approximation, and better numerical results can be obtained if more factors are included into calculation. Thus, in this chapter factors that are out of the scope of the study should be summarised, such as:

  • cooling energy, (what about use of split air conditioners?)
  • aspects that remain constant in all the compared scenarios eg. orientation or shading by adjoining buildings,
  • increased use of energy based on artificial lighting, electric vehicular charges, use of energy efficient lighting such as LEDs, use of lighting control strategies
  • human factors, social or administrative nature,
  • historic buildings conditioning, etc.

The a.m. factors can be addressed in more detail at more advanced levels of the study.

Second, it would be good to list what aspects of the model require inputting local data when applied to other cities in Europe. I do not agree that some particular aspects of Spanish context could limit the application of this method. I understand that the method is universal, but local aspects and local data should be used for calculations in other locations. Such local aspects cover local residential buildings and appartements characteristics, local building codes and their changes in the past, local technical codes, local weather conditions to name a few.

In Conclusions I would stress at universal aspects of the model that should work well in other European countries, i.e. using cadastral data in order to calculate the relevant physical characteristics of the building. Also, it is important that you methodology is flexible, i.e. allows for defining various (local) scenarios for renovation and for taking into account changes of daily habits due to various reasons (i.e. not only related to pandemic).

Round 3

Reviewer 2 Report

Thank you for considering my suggestions.